# Applications of *Bacillus subtilis* Protein Display for Medicine, Catalysis, Environmental Remediation, and Protein Engineering

**DOI:** 10.3390/microorganisms12010097

**Published:** 2024-01-03

**Authors:** Asieh Mahmoodi, Edgardo T. Farinas

**Affiliations:** New Jersey Institute of Technology, Newark, NJ 07102, USA; am289@njit.edu

**Keywords:** protein display, *Bacillus subtilis* spore, protein stability, enzyme activity

## Abstract

*Bacillus subtilis* spores offer several advantages that make them attractive for protein display. For example, protein folding issues associated with unfolded polypeptide chains crossing membranes are circumvented. In addition, they can withstand physical and chemical extremes such as heat, desiccation, radiation, ultraviolet light, and oxidizing agents. As a result, the sequence of the displayed protein can be easily obtained even under harsh screening conditions. Next, immobilized proteins have many economic and technological advantages. They can be easily separated from the reaction and the protein stability is increased in harsh environments. In traditional immobilization methods, proteins are expressed and purified and then they are attached to a matrix. In contrast, immobilization occurs naturally during the sporulation process. They can be easily separated from the reaction and the protein stability is increased in harsh environments. Spores are also amenable to high-throughput screening for protein engineering and optimization. Furthermore, they can be used in a wide array of biotechnological and industrial applications such as vaccines, bioabsorbants to remove toxic chemicals, whole-cell catalysts, bioremediation, and biosensors. Lastly, spores are easily produced in large quantities, have a good safety record, and can be used as additives in foods and drugs.

## 1. Introduction

### 1.1. Protein Display

The 2018 Nobel Prize was shared between George Smith and Gregory P. Winter for phage display of peptides and antibodies and Frances Arnold for directed evolution of proteins. The award recognized that evolution can be mimicked in the laboratory to design and optimize proteins to solve chemical problems. The goal of protein engineering and optimization is to create proteins designed to order. These strategies have emerged as the method of choice for improving protein properties and as a valuable tool for investigating fundamental structure/function relationships [1]. Laboratory evolution is an iterative process of creating gene libraries by random mutagenesis or recombination followed by a screen or selection to identify mutants with the desired properties. The process involves iterations of the following steps (Figure 1): (1) Gene libraries are generated from a parent by random mutagenesis or recombination. (2) The libraries are inserted into an expression vector. (3) The vectors are transformed into host cells (one cell, one mutant). (4) The cells are screened for the protein property of interest. (5) The improved gene(s) are isolated, and the process repeated. The most important step in a directed evolution experiment is a robust high-throughput screen (HTS) to identify active variants from large protein libraries, which can be between 10^5^–10^8^ variants.

Phage and microbial cell-surface display has been a successful method for screening protein libraries (Figure 2) [2,3]. However, the bottleneck oftentimes is the screen or selection. Screening protein libraries on the cell surface is more convenient than assaying for the activity of recombinant proteins that remain inside of the cell. For example, substrates or binding partners are freely accessible to the target protein. This contrasts with assaying proteins that remain within the cell because the screening substrate may not be freely diffusible across the membranes. As a result, the screening procedure requires microtiter plate assays necessitating multiple steps to measure protein function such as liquid handling, cell disruption, and centrifugation. Alternatively, protein display does not require multistep liquid handling. Going on, the assay conditions are easier to control with displayed proteins. For instance, cell lysis may be required for proteins expressed inside a cell and the assay environment will contain the contents of the lysed cells. For protein display, the cells can be centrifuged and resuspended in the desired medium. Screening for protein/substrate or protein/protein interactions is more convenient (Figure 2). A gene library is constructed (1) cloned into a plasmid (2). Next, the cells will express a unique member of the library (3) and protein binding or enzymatic activity is assayed (4). The cells are washed to remove the cells that do not bind (5) and the cells that bind are freed and collected (6) and allowed to grow (7). Finally, the gene is isolated, and another iteration can be performed.

Displayed proteins are amenable to high-throughput screening methods using tools such as flow cytometry. Fluorescence-activated cell sorting (FACS) based screening was originally developed for protein-ligand interactions and applied to enzyme screening [4,5,6]. FACS screening can routinely analyze and quantify 100,000 clones/second, and the rare cells expressing the improved protein can be sorted from the population. Consequently, the sorted cells can be cultivated, and the sequence of the gene retrieved.

Although microbe cell-surface protein display appears ideal for screening purposes, shortcomings are still evident. A major concern of protein display is the compatibility between the host and the displayed protein. Successful display necessitates that the anchor structure: (1) contains a signal peptide or transporting signal allowing the premature fusion protein to pass through the inner membrane; (2) keeps the fusion protein attached to the cell surface; (3) allows foreign sequences to be fused; (4) resists protease degradation. In short, the underlying issue relates to protein folding concerns. For example, the proteins expressed in gram-negative bacteria such as *Escherichia coli* are synthesized in the reducing environment of the cytoplasm. Hence, the formation of the correct disulfide bonds in proteins, such as antibodies, is disfavored resulting in misfolded proteins. Protein folding issues are further magnified in *E. coli* because the protein expressed in the cytoplasm must first cross the plasma membrane, followed by passage through the peptidoglycan layer, and finally traversing the outer membrane. Some of these issues have been partially addressed by using mutant strains having relatively high oxidizing cytoplasm [7] or displaying proteins on the outer surface of the periplasm surface display for yeast and phage (Figure 3A). 

Further limitations are associated with FACS screening of surfaced displayed proteins in microorganisms such as *E. coli*. The cell viability of the sorted cells can be an issue when determining the DNA sequence of the improved protein (Figure 3B). In commercially available flow cytometers, the positive cells are sorted into a collection tube, and the cells are recovered by filtration through a membrane. Next, the membrane is transferred to an agar plate and incubated until colonies are developed. Alternatively, the sorted cell can be directly cultured in liquid media. Finally, the sequence that codes for the protein can be identified. Cells containing the improved protein can be sorted, but they cannot be cultivated and isolated because they are no longer viable. This can occur with harsh screening conditions or toxic substrates. As a result, the DNA sequence cannot be identified for the improved protein resulting in the possibility of missing the mutant.

### 1.2. Advantages of Spore Display

Displaying recombinant proteins on *B. subtilis* (a gram-positive bacterium) spores offer several advantages. Protein folding issues associated with unfolded polypeptide chains crossing membranes are circumvented. Briefly, the sporulation of *B. subtilis* proceeds through a well-defined series of morphological and biosynthetic steps (Figure 4) [8,9]. Sporulation is initiated upon starvation which results in an inactive endospore (spore). The onset of sporulation is characterized by the growing cell forming a large and small compartment, which is known as the mother cell and forespore, respectively. The forespore develops into a spore, and the role of the mother cell is to nurture the forespore until it is fully matured. Nearing the end of the sporulation cycle, proteins synthesized in the mother cell are assembled around the forespore to construct the outer coat. In the final step, the mother cell disrupts releasing the fully formed spore. 

In contrast to other display systems, proteins found on the spore outer coat do not travel across membranes. Furthermore, the chaperones present in the mother cell are available to assist in protein folding. In short, spore display offers the opportunity to access a region of protein space that was previously difficult to express with other display platforms.

Spores have additional properties making them attractive for surface-displayed proteins. They can withstand many physical and chemical extremes such as heat, desiccation, radiation, ultraviolet light, and oxidizing agents [10]. It is believed that spores can remain viable for more than a million years [10,11]. Hence, the sequence of the displayed protein can be easily obtained even under harsh screening conditions. On the other hand, vegetative cells cannot endure such treatments. Taking advantage of the resistant properties of spores, displayed proteins are “preimmobilized”. In traditional immobilization methods, proteins are expressed and purified and then they are attached to a matrix. In contrast, immobilization occurs naturally during the sporulation process. In general, immobilized proteins have many economic and technological advantages [12]. They can be easily separated from the reaction and the protein stability is increased in harsh environments. Spores are also amenable to high-throughput screening using FACS, and it has been demonstrated that a displayed protein can be analyzed using this technology [13]. Furthermore, they can be used in a wide array of biotechnological and industrial applications such as vaccines, bioabsorbants to remove toxic chemicals, whole-cell catalysts, and biosensors [5]. Lastly, spores are easily produced in large quantities (Table 1).

*B. subtilis* provides a convenient system to explore spore display. An advantage of using *B. subtilis* is that the spore structure and sporulation cycle have been well studied [8,9,41]. Furthermore, the ease of using genetic tools and genomic data facilitates the construction of recombinant *B. subtilis* spores [42,43]. Finally, they have a good safety record for humans and livestock and can be used as additives in foods and drugs [44,45]. 

As mentioned already, spores are highly resistant to harsh conditions. This positive trait can also have negative effects on heterologous protein display with spores. Electron and atomic force microscopy investigations demonstrated that stable macrostructures could form. This arises from the self-assembly CotY, CotE, CotV and CotW to create one-dimensional fibers, two-dimensional sheets, and three-dimensional stacks. This feature contributes to the inert spore property. This stable structure may hinder the design of successful heterologous protein/spore coat protein fusions [46].

### 1.3. B. subtilis Anchor Proteins for Protein Display

The *B subtilis* spore coat provides a protective structure. The coat is multilayered, and it contains more the 70 distinct proteins. The architecture of the spore begins with the core, which contains the genetic information (DNA), and it is surrounded by the cortex. The following levels are the basement, inner coat, and outer coat layers. The final layer is the crust (Figure 5)

The spore-displayed protein is fused to an endogenous protein and there are several candidates. Several spore coat proteins were evaluated for display, which used green fluorescent protein (GFP) and two distinct laccases. They were fused at the N- and C- terminus to 6 crust proteins, which resulted in 12 vectors [47]. The crust proteins investigated were CotV, CotW, CotX, CotY, CotZ, and CgeA. GFP-display demonstrated that N-terminal fusions were more effective than C-terminal fusions. It was found that N-terminus fusions of CotX, CotY, and CotZ showed the greatest fluorescence. It was estimated that each spore contained approximately 16,000–20,000 molecules. On the other hand, 10,000 and 4000 molecules per spore were estimated for CotV and CotW, respectively. GFP-CgeA had very little functional expression of GFP with only 130 molecules per spore. The best C-terminal fusion was for CotY with approximately 17,000 molecules per spore. GFP-CotZ C-terminal fusion showed about 5000 GFG, which is about 4 times lower than the N-terminal fusion. The coat proteins above were also used to fuse laccases from *Escherichia coli* (EcoL) and *Bacillus pumilus* (BpuL). In addition, CotB, CotC, and CotG have also been used as anchor proteins for display. These proteins are tyrosine-rich, multimers, and found on the outer coat [48,49,50,51,52].

## 2. Applications

Spores are amenable to a variety of applications in vaccine development, biocatalysis, bioremediation, protein engineering efforts, and drug delivery. This is due to the spore properties mentioned above.

### 2.1. Vaccine Development/Drug Delivery

Protein spore display have been utilized to present antigens and elicit immune responses [5,14,53]. The proof of concept was demonstrated by fusing a C-terminal tetanus fragment (TTFC) to CotB (TTFC-CotB) [13]. Display was confirmed by western blot and fluorescent-activated cell sorting (FACS). Consequently, the TTFC-CotB was later shown that intranasal dosage produced a mucosal (IgA) and systemic (IgG) reaction in murine models [15]. In another report, spore displayed TTFC-CotB were treated intragastrically, and high serum TTFC-specific antibodies were induced towards the antigen [54]. In a related investigation, CotC was used as an anchor for TTFC and heat-labile toxin of *E. coli* (LTB) [55]. These antigens were administered orally and injected intraperitoneal in mice. In both instances, an immune response was evoked. 

Vaccines have proven to be effective for the prevention of COVID-19 infection. A subset of people fear needles and experience side effects from current vaccines. Hence, an oral dose may be a suitable alternative. The receptor binding domain (RBD) of the SARS-CoV-2 spike glycoprotein was fused to CotY or CotZ and was proposed to be used as an oral vaccine for the SARS-CoV-2 virus [16]. In another investigation, the spike protein of the ancestorial SARS-CoV-2 coronavirus was fused to CotC. This fusion with additional adjuvants demonstrated activation of macrophages and dendritic cells [17]. 

The feedlot industry suffers significant health issues that arise from bovine respiratory disease (BRD) [56]. Currently, antibiotics have been used to prevent BRD because there is a deficiency of successful vaccines. As a result, a mucosal vaccine was developed for *Mannheimia haemolytica*, which is a BRD pathogen. *B. subtilis* spores were utilized as an adjuvant. A chimeric protein (MhCP) was constructed that was composed of the neutralizing epitopes from *M. haemolytica leukotoxin* A (NLKT) and the outer membrane protein PlpE. The conditions were optimized for the adsorption of MhCP to *B. subtilis* spores. The spore-bound antigen (MhCP-spore) was administered to mice through intranasal and intragastric routes and was found to be more successful compared with free MhCP. Intranasal was found to be the most effective in eliciting the greatest IgG response [18]. Next, another potential mucosal vaccine was developed for porcine circovirus type 2 (PCV2). The PCV2 capsid protein (Cap) was fused to CotB, and fusion stimulated robust humoral and mucosal immune responses [57].

*B. subtilis* spores are amenable to orally delivering drugs. For example, the spore outer coat (spore) was covalently linked to curcumin (Cur) and folate (FA) to create Cur-FA-spore. This species was developed for therapy against colon cancer. Cur-FA-spore was delivered to the colon, and the drug was released by crossing the gastric barrier. Results demonstrated that Spore-Cur-FA improves oral bioavailability of Cur, which inhibits colon cancer cells [19]. In short, spores have been demonstrated to be an additional tool for vaccine delivery.

### 2.2. Biocatalysis

Protein spore display has been utilized for robust biocatalysis. Spores displayed enzymes have been constructed to meet the industrial conditions, which require stability in a variety of conditions. For example, β-galactosidase (β-Gal) was displayed and used for transgalactosylation in biphasic reaction mixtures [20,21]. β-Gal was fused to the coat protein CotG and the construct was evaluated in a variety of organic solvents, which included *n*-hexane, ethyl ether, toluene, ethyl acetate, acetonitrile, and ethanol. These solvents had log *p* values or solvent hydrophobicity ranging from 3.5 to −0.24. In addition, the β-Gal-CotG fusion resulted in increased thermostability compared to the free enzyme. Furthermore, the β-Gal-CotG spores were treated with glutaraldehyde for chemical cross-linking, and the enzyme was further stabilized. β-Gal-CotG spores synthesized octyl-D-galactopyranoside with yields up to 8.1 g/L (27.7 mM) with lactose (100 mM) and octanol (100 mM) in a solvent mixture of phosphate buffer and ethyl ether. Catalysis occurred at the interface between the two solvents. This was a demonstration that spores with a displayed enzyme can be used as a phase transfer biocatalysis. In another investigation, the functional expression of β-Gal was evaluated with fusions to the crust proteins, CotC, CotY, and CotZ. Displayed proteins showed enzyme activity, which demonstrates that these coat proteins are suitable candidates for enzyme display [22].

D-psicose 3-epimerase (DPEase) is useful for d-allulose synthesis [23]. D-allulose is valuable in the food, pharmaceutical, and healthcare industries. They induce physiological responses, which include antiobesity, antihyperglycemic, antidiabetic, anti-inflammatory, antioxidant, and neuroprotective effects [58,59]. The enzyme was fused to the C-terminus of CotZ. DPEase-CotZ showed optimal temperature and pH at 55 °C and 7–5–8.0, respectively. The biocatalyst (30 g/L spores) yielded d-allulose (85 g/L) from fructose (500 g/L d-fructose) after 12 h.

Lipases have also been displayed and they catalyze the hydrolysis of fats. They are industrially versatile and used in food, detergent, and pharmaceutical industries. [24,25,60,61,62,63] *Thermotoga maritimas* lipase, TM1350, was fused to CotB, and displayed enzyme had an optimal temperature and pH of 80 °C and 9, respectively. The nonimmobilized TM1350 had an optimal temperature and pH of 70 °C and 7.5, respectively. [24] In addition, displayed TM1350 retained 18% higher activity. Furthermore, the TN1350-CotB was able to be recycled without a significant decrease in activity. 

Esterases catalyze ester hydrolysis and are useful in the food industry for flavor enhancement, pharmaceutical synthesis, and bioremediation to name a few examples. An esterase was fused to CotB and it had a temperature optimum of 60 °C and retained 70% of the original activity after 5 h. In addition, Esterase-CotB maintained 65.2% activity dimethyl sulfoxide (20% *v*/*v*) for 7 h [25]. 

Additional industrial applications include spore-displayed *Acetobacter pasteurianus* AdhA (alcohol dehydrogenase). They catalyze the interconversion between alcohols and ketones with the reduction of nicotinamide adenine dinucleotide (NAD^+^) to NADH. They have been used for improved tolerance toward ethanol for improved flavor for liqueur production [26]. Next, spore surface-displayed N-acetyl-D-neuraminic acid aldolase was utilized to synthesize N-acetyl-D-neuraminic from N-acetyl-D-glucosamine. The enzyme was fused to CotG, and the product serves a variety of biological roles and may have medical applications [27]. 

Nitrilases catalyze the hydrolysis of nitriles to amino and carboxylic acid [64]. These enzymes transform toxic nitrile compounds into benign and valuable acids with the production of ammonia. The free nitrilase from *Thermotoga maritima* MSB8, a hyperthermophilic bacterium, has a pH and temperature optimum of 7.5 and 45 °C, respectively. The nitrilase was fused to CotG and the pH and temperature optimum were determined to be 8.0 and 50 °C, respectively. In addition, the displayed enzyme was incubated at 75 °C at pH 8.0 for 1 h and the fusion was found to have improved thermal and pH stability [28].

Spore-displayed enzymes have shown beneficial properties for industrial use. The thermostability and tolerance to organic solvents are enhanced for the fusion. In addition, they are easy to separate from the reaction mixtures and can be recycled.

### 2.3. Protein Engineering and Optimization

The *B. subtilus* coat protein CotC is an enzyme with laccase activity, which is in the oxidoreductase family. This enzyme is considered a “generalist”. It catalyzes a wide range of substrates by reducing O_2_ to H_2_O with the concomitant generation of radicals or other reactive intermediates [65]. This feature is attractive for a variety of industrial and biotechnological applications in environmental science, bioremediation, and biofuel production [64,66,67]. The wild-type CotC has been engineered and optimized by directed evolution for substrate specificity, organic solvent stability, and pH stability. The general strategy was to create a mutant library. Then, the library was integrated into the genome into the non-essential *amyE* gene by double crossover recombination. The resulting transformants were sporulated and the library, which contained approximately 3000 variants, was expressed in the spore coat. The fittest variants were expressed, purified, and characterized to confirm the enhanced trait. First, the substrate specificity was narrowed for wild-type CotA [29,30,31]. The enzyme is active towards ABTS [diammonium 2,2 (azino-bis(3-ethylbenzothiazoline-6-sulfonate)] and SGZ (4-hydroxy-3,5-dimethoxy-benzaldehyde azine). A mutant CotA was found to be 120-fold more specific for ABTS. A saturation mutagenesis library was constructed that targeted all 23 amino acids. These amino acids were based on the ABTS-CotA crystal structure [65]. The chosen amino acids are within 6 A˚ of the bound substrate. This was the first demonstration that *B. subtilus* spores could be a vehicle for directed evolution protein engineering. Next, spores can remain viable under harsh conditions and this property was explored to determine if spores can be utilized to engineer enzymes under extreme conditions such as high organic solvent concentration [32]. The library was constructed by error-prone PCR and the library was assayed in 60% dimethyl sulfoxide. A Thr480Ala variant was identified to be 2.4-fold more active than wild-type. The variant was more active than wild-type in various DMSO concentrations ranging from 0–70%. In addition, polar protic (ethanol and methanol) and polar aprotic (acetonitrile) organic solvents were evaluated. The variant was more active in all solvents assessed. This study demonstrated that spores can be used to engineer proteins with extreme properties. Finally, the pH optimum was 4 for the wild-type [33]. However, the half-life (*t*_1/2_) was only 50.9 min. An error-prone PCR library was constructed and a Glu498Gly amino acid substitution was identified. The *t*_1/2_ was increased to 1264 min. Then, the addition Thr480Ala, which was found for organic solvent stability, was used to construct the Thr480Ala/Glu498Gly variant, and the *t*_1/2_ was increased to 3166 min. In a final investigation, Thr480Ala expressed on the spore coat was evaluated as an effective biocatalysis for oxidation of a variety of phenolic substrates, (+)-catechin, (−)-epicatechin, and sinapic acid in various aprotic and protic organic solvents [34]. In all cases, the *V*_max_/K_m_ for Thr480Ala was greater than the wild-type. The variant retained approximately 60% activity for (+)-catechin when it was recycled 7 times in 23 h. In addition, the variant had a total product yield that was 3.1-fold greater than the wild-type. In short, spores can be used for protein engineering and optimization for enzymes with extreme properties.

### 2.4. Environmental Applications 

Enzyme-displayed spores have also been used for bioremediation by taking advantage of the properties of spores. Chitinase hydrolyzes random endo-hydrolysis of N-acetyl-β-D-glucosaminide (1→4)-β-linkages in chitin and chitodextrins. Applications include fertilizer production, biomaterial synthesis, and enhancement of fungicides and insecticides [68]. Chitinase was fused to CotG and was demonstrated to inhibit fungi (*Rhizoctonia solani* and *Trichoderma harzianum*) [69]. 

Tyrosinase can be used to remediate phenol-polluted environments. It was anchored to CotE (Tyr-CotE) and the activity was monitored for L-tyrosine. Tyr-CotE was maintained in water at room temperature for 15 days without a significant decrease in activity. In addition, Tyr-CotE retained 62% activity after six washing cycles [35]. Another example is the bioremediation of atrazine, which is chlorinated triazine. It is used in the agriculture industry to prevent broadleaf weed growth in crops such as soybeans, corn, and sugarcane, and it is also harmful to the human and animal endocrine systems. A chlorohydrolase was fused to BclA N-terminal targeting and attachment sequence of attachment domain of the BclA spore surface nap layer protein and expressed in *B. thuringiensis*. It was demonstrated that the fusion catalyzes the degradation of atrazine [70]. Another chlorinated hazard, sulfur mustard (2,2′-dichlorethyl sulfide) is a target for bioremediation by haloalkane dehalogenase (DhaA). However, DhaA is not stable under harsh environments, which limits its potential. Hence, DhaA was fused to CotG, and this was the first report of spore display of Dha and it was assayed with sulfur mustard analog (2-chloroethyl ethylsulfide). The displayed enzyme remained active, which demonstrates that DhaA can be used for the remediation of contaminated environments [36].

Microbial metabolic pathway products that result from *meta* -cleavage of aromatic compounds such as catechols and polychlorinated biphenyl are HODAs (2-hydroxy-6-oxohexa-2,4-dienoic acids), which arise from *meta*-cleavage [71,72] It has been shown that HODAs accumulate and hinder aromatic mineralization. A *meta*-cleavage product (MCP) hydrolase (MfphA and BphD) was fused to CotG. The optimal pH and temperature were determined to be pH 7 at 70 °C. It was also found that the fusions remained stable at 80 °C at pH 10 and retained approximately 80% activity. In addition, the fusions can be recycled up to ten times without significant loss of activity. Spore-displayed enzymes have applications in HODAs transformation [37]. 

Metal ion toxicity is an issue that requires attention. Nickel is found extensively in the environment, and it is an essential trace transition metal for animals and human beings. However, it is also an environmental pollutant and can cause cardiovascular and kidney disease, lung scarring, and cancer [73]. CotB was fused to eighteen histidine residues. The fusion was capable of binding nickel statistically higher than spores alone in a pH range between 5–9. The optimum conditions were pH 7, 25 °C, and 25 mg spores. The pH and temperature did significantly affect the absorption [38]. Next, rare-earth elements (REE) are used for common day devices such as smartphones, computer hard disks, televisions, and other electronic displays. REEs are very scarce and large-scale mining is required that uses strong acids. As a result, toxic chemicals are released. It has been shown that the REEs Dy^3+^ and Tb^3+^ accumulate in the outer layers of the spore coat. The REEs are released upon germination. The prospect is that spores can be used as an absorbent for REEs [39].

*B. subtilis* spores have been used to monitor arsenic. Arsenic (As) is very hazardous because of its high occurrence in water and soil. Hence, it enters the food chain and results in gastrointestinal issues, cancer, and arsenicosis. This is a worldwide issue, and it affects about 140 million people in 50 countries. *B. subtilis* is transformed with a plasmid that is under the control of the concentration of As(III) with a green fluorescent protein reporter (GFP). The spores express GFP in the presence of As(III) in a dynamic range from 0.1 to 1000 μM 4 h after the beginning of germination. In addition, it is specific and sensitive towards As. This may be a sensor that can be used directly for environmental samples [40,74].

## 3. Conclusions

Spores displayed proteins are versatile and provide a platform for several applications. They can express a diverse assortment of proteins and can be used to develop applications in biotechnology, medicine, agriculture, and protein design and engineering. This is possible because they resist extreme conditions such as heat, chemicals, and radiation. Next, they remain stable and have a long shelf life. Going on, they are easily produced in large quantities and molecular biology tools are readily available. In addition, *B. subtilis* spores are biologically safe and can be administered without harm. In short, spores are a complementary tool for current traditional protein display technologies, such as *E. coli*, yeast, and phage. For example, spores can express a region of protein space that traditional display technologies cannot access and improve the properties of the displayed protein.

## Figures and Tables

**Figure 1 microorganisms-12-00097-f001:**
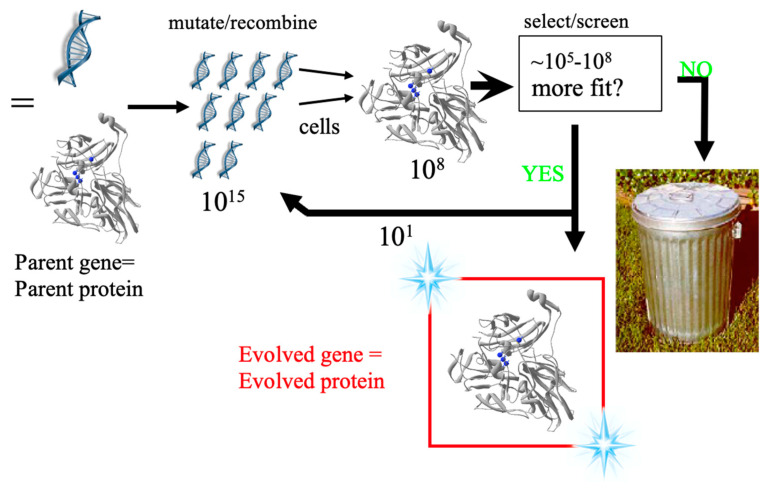
Directed evolution cycle.

**Figure 2 microorganisms-12-00097-f002:**
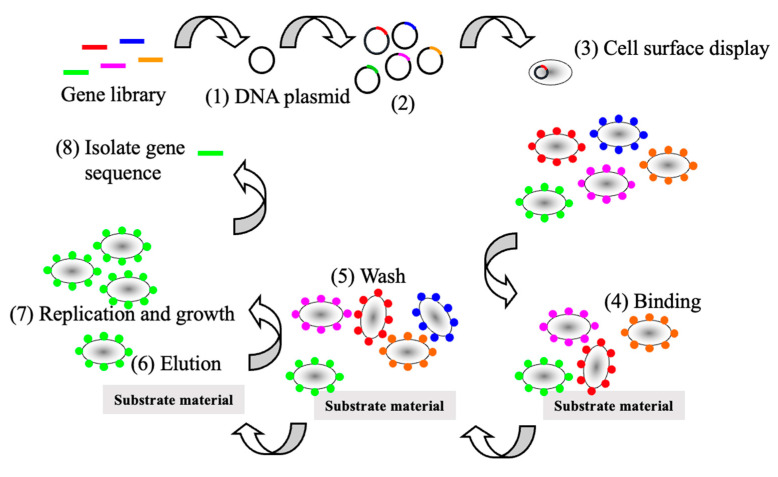
Screening protein displayed libraries.

**Figure 3 microorganisms-12-00097-f003:**
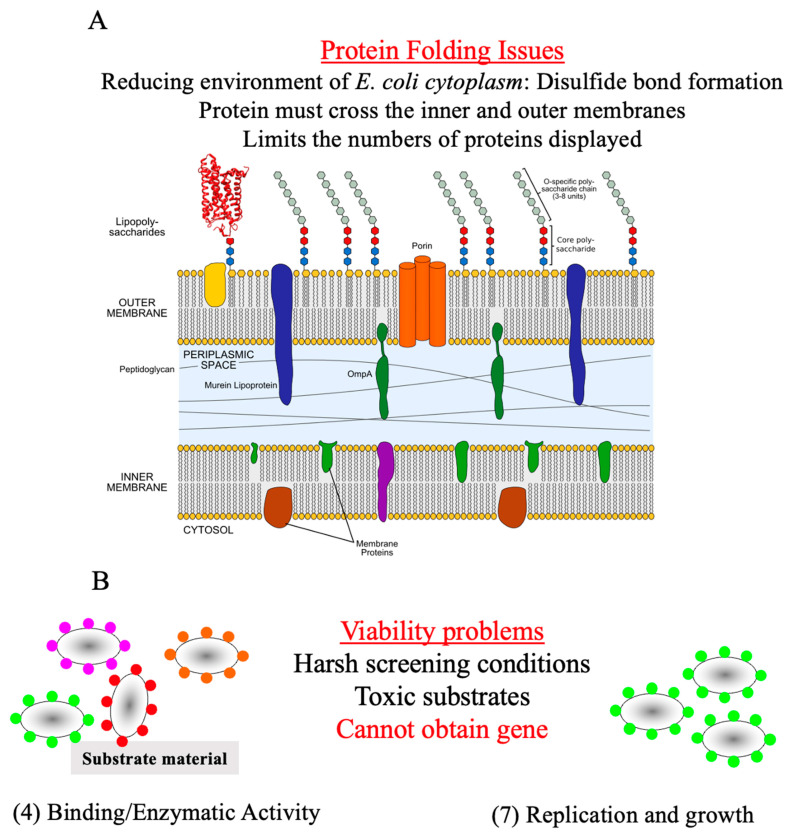
(**A**). Protein folding concerns for protein display. (**B**). Viability issues that prevent recovery of the gene.

**Figure 4 microorganisms-12-00097-f004:**
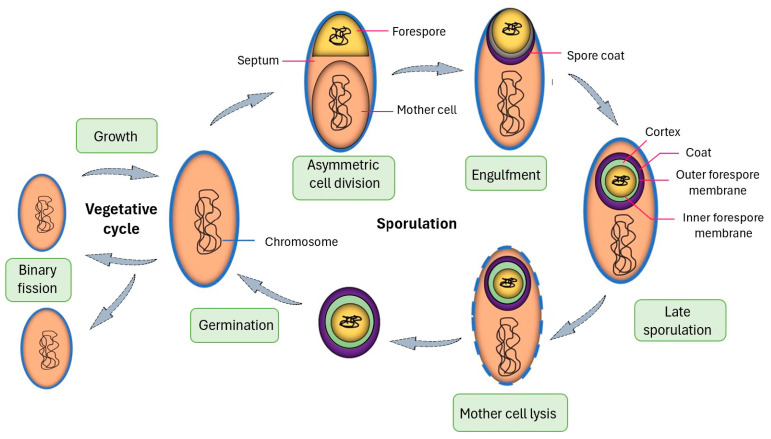
*Bacillus subtilis* sporulation cycle.

**Figure 5 microorganisms-12-00097-f005:**
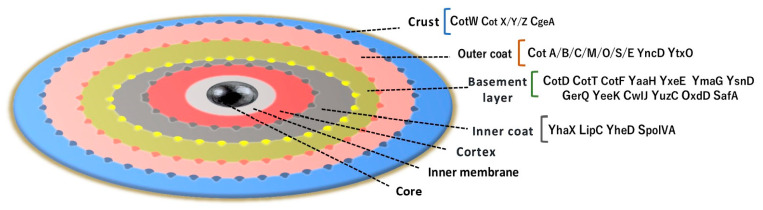
Spore architecture.

**Table 1 microorganisms-12-00097-t001:** Spore applications.

Application	Anchor	Fusion	Function	References
Vaccine; adjuvant; drug delivery	CotB	C-terminal tetanus fragment (TTFC)	Intranasal dosage produced a mucosal (IgA) and systemic (IgG) reaction in murine mod	[13,14,15]
	CotZ, CotY	SARS-CoV-2 Receptor binding domain	Oral vaccine against SARS-CoV-2 virus	[16,17]
	Spore	Neutralizing epitopes from *Mannheimia haemolytica*	Adjuvant for bovine respiratory disease	[18]
	Spore	Covalent linkage of curcumin and folate	Drug delivery for colon cancer	[19]
Biocatalysis	CotG	β-galactosidase	Transgalactosylation in biphasic reaction mixtures	[20,21]
	CotC, CotY, CotZ	β-galactosidase	β-gal reaction; evaluate as potential enzymatic anchors	[22]
	CotZ	D-psicose 3-epimerase	Allulose synthesis	[23]
	CotB	Lipase	Demonstrates advantage of enzyme displayed enzyme comparted to non-immobilized biocatalysis	[24]
	CotB	Esterase	Demonstrates advantage of enzyme displayed enzyme comparted to non-immobilized biocatalysis	[25]
	CotC	Alcohol dehydrogenase	Ethanol tolerance for flavor production in liquor	[26]
	CotG	N-acetyl-D-neuraminic acid aldolase	Synthesis of N-acetyl-D-neuraminic from N-acetyl-D-glucosamine	[27]
	CotG	Nitrilase	Hydrolysis of nitriles to ammonia and carboxylic acid	[28]
	CotC	Genomic substitution of wild-type CotC (laccase)		[29,30,31,32,33,34]
Environmental applications	CotE	Tyrosinase	Phenol polluted environments	[35]
	CotG	Haloalkane dehalogenase	Degradation of sulfur mustard	[36]
	CotG	*Meta*-cleavage product (MCP) hydrolase (MfphA and BphD)	2-hydroxy-6-oxohexa-2,4-dienoic acids transformation	[37]
	CotB	His18	Nickel binding	[38]
	Spore	Not applicable	Rare earth element binding	[39]
	Spore	Not applicable	Arsenic sensor	[40]

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
