# Peer review of "Applications of Bacillus subtilis Protein Display for Medicine, Catalysis, Environmental Remediation, and Protein Engineering"

_microorganisms, 2024, doi:10.3390/microorganisms12010097_

Round 1

Reviewer 1 Report

Comments and Suggestions for Authors

The main topic of this review was to give an overview of the use of spores for protein display, biocatalysis, and protein engineering as well as the therapeutic potential of spores as probiotics, vaccine vehicles, and drug delivery systems. Different parts of the review largely differ in the quality of presented information. The weaknesses of the review are described in major points. There are additional suggestions for changes and typos pointed out in minor points.

Major comments

1.     Figure 3. “Protein must cross several membranes”. E. coli has only two membranes.

2.     Line 120: “In short, spore display offers the opportunity to access a region of protein space that was previously difficult to express with other display platforms.” It is important also to describe the disadvantages of protein spore displays. Among them is the fact that the structure of some, especially, crust spore coat proteins forms two-dimensional crystals, wires, etc. These represent very crowded structures for we have no 3D structures of the coat proteins. This makes it difficult to design a specific site of coat protein for fusion with the protein of interest. Subsequently, improper fusion can hinder fully or partially the function of the protein of interest.

3.     Redrew Figure 4 as cell 3 is scratched with two membranes but not cells 4 and 5. This is not true.

4.     Redrew Figure 5 as it is not true how it is that the crust is made only of protein CotW and describe also what is in the middle of the spore.

5.     Line 233: “Crust proteins showed enzyme activity, which…”. It is not true because the crust proteins have no enzyme activity.

Minor comments

1.     Line 113: The sentence “…and the role of the mother cell is to nurture the spore until it is fully matured.” should be “…and the role of the mother cell is to nurture the forespore until it is fully matured.”

2.     Line 127: The sentence “It is believed that spores can remain viable for up to several thousand years” should be It is believed that spores can remain viable for more than million of years”.

3.     Line 290: “These amino acids were based on the ABTS-CotA crystal structure.” This statement needs citation.

Author Response

  1. Figure 3. “Protein must cross several membranes”. E. coli has only two membranes.

Figure 3 was corrected from” Protein must cross several membranes” to “Protein must cross the inner and outer membranes”

  1. Line 120: “In short, spore display offers the opportunity to access a region of protein space that was previously difficult to express with other display platforms.” It is important also to describe the disadvantages of protein spore displays. Among them is the fact that the structure of some, especially, crust spore coat proteins forms two-dimensional crystals, wires, etc. These represent very crowded structures for we have no 3D structures of the coat proteins. This makes it difficult to design a specific site of coat protein for fusion with the protein of interest. Subsequently, improper fusion can hinder fully or partially the function of the protein of interest.

We agree with the Reviewer !.

This paragraph was added to the manuscript at line 107 of the original manuscript.  The B. subtilis spore coat is made up of at least 80 proteins. Some of these proteins have a two-dimensional crystal structure and only a fraction of these proteins have been solved by electron microscopy. These concerns pose a challenge in choosing a protein for the fusion of heterologous proteins fusion to coat proteins for display. There has been a recent investigation to identify suitable coat proteins for display fusions (vide infra).

  1. Redrew Figure 4 as cell 3 is scratched with two membranes but not cells 4 and 5. This is not true. 

      Figure 4 describing sporulation was redone with additional details.

  1. Redrew Figure 5 as it is not true how it is that the crust is made only of protein CotW and describe also what is in the middle of the spore

Figure 5 was redone.

  1. Line 233: “Crust proteins showed enzyme activity, which…”. It is not true because the crust proteins have no enzyme activity.

Changed “Crust proteins showed enzyme activity, which…” to “Displayed proteins showed enzymatic activity, which…”

Minor comments

  1. Line 113: The sentence “…and the role of the mother cell is to nurture the spore until it is fully matured.” should be “…and the role of the mother cell is to nurture the forespore until it is fully matured.”

Corrected as suggested.

  1. Line 127: The sentence “It is believed that spores can remain viable for up to several thousand years” should be “It is believed that spores can remain viable for more than million of years”.

Corrected as suggested.

  1. Line 290: “These amino acids were based on the ABTS-CotA crystal structure.” This statement needs citation.

Reference 53 is the citation. “These amino acids were based on the ABTS-CotA crystal structure.” Was changed to “These amino acids were based on the ABTS-CotA crystal structure. [53]”

Reviewer 2 Report

Comments and Suggestions for Authors

This manuscript foucs on Bacillus. subtilis spores, and introduces the advantages of spore display, further summaries the different applications of spore display. Thus, this review is meanful and helpful to the field study on pore protein display. However, this review still needs some revision before acceptance for publication.

My detailed comments are as follows:

1.     Why B. subtilis have a good safety record and can be used as additives in foods and drugs?

2.     Some spore coat proteins are not exposed in the outside of spore, how they can be used as an anchor (such as CotB) is fused a antigen to trigger a inmmune response?

3.     The content of “2.5 Biomonitoring” seem too thin compared with other parts, suggest authors to find more advances of biomonitoring about using spore.

4.     Literatures cited in this review seems too old to reflect the recent advances of this reaseach field.

Author Response

  1. Why B. subtilis have a good safety record and can be used as additives in foods and drugs?

References 17 and 18 were added to the text.

  1. Cutting, S.M., Bacillus probiotics. Food Microbiology, 2011. 28(2): p. 214-220.
  2. Hong, H.A., et al., The safety of Bacillus subtilis and Bacillus indicus as food probiotics. Journal of Applied Microbiology, 2008. 105(2): p. 510-520.

  1. Some spore coat proteins are not exposed in the outside of spore, how they can be used as an anchor (such as CotB) is fuse an antigen to trigger an immune response?

I agree with Reviewer 2.  However, the experimental data is convincing that an immune response was elicited. 

  1. The content of “2.5 Biomonitoring” seem too thin compared with other parts, suggest authors to find more advances of biomonitoring about using spore.

Section 2.4 was renamed from Bioremediation to Environmental Applications.  Section 2.5 was incorporated into the newly named section.

  1. Literatures cited in this review seems too old to reflect the recent advances of this research field. 

The review was written in the historical perspective of the 2018 Nobel Prize.  This included protein engineering, monitoring, evaluation of spore coat proteins suitable for heterologous protein fusions, and the use of SARS-CoV-2

Round 2

Reviewer 1 Report

Comments and Suggestions for Authors

In general, the authors made most of the suggested changes. However, one advised change is not fulfilled at all (lines 108-117). This added new text is just wrong. First of all, part of the added text is repeated twice. Secondly, what is (vide infra)” at the end of the paragraph? Thirdly, the citation [8] is just not right. It is a review from 2009 but there are no two-dimensional crystal structures of coat protein mentioned there. The TEM supramolecular structures formed by self-assembling coat proteins were first published in 2015 in Molecular Microbiology. In addition, the last sentence of the paragraph “There has been a recent investigation to identify suitable coat proteins for display fusions.” requires citation/s.

Author Response

Track changes have been enabled to aid the review of the changes. We thank the reviewer for the comments.

In general, the authors made most of the suggested changes. However, one advised change is not fulfilled at all (lines 108-117). This added new text is just wrong.

(1) First of all, part of the added text is repeated twice.

REPLY

The repeated text has been removed.

(2) Secondly, what is “ (vide infra)” at the end of the paragraph?

REPLY

This was removed from the deletion mentioned above.

(3)Thirdly, the citation [8] is just not right. It is a review from 2009 but there are no two-dimensional crystal structures of coat protein mentioned there. The TEM supramolecular structures formed by self-assembling coat proteins were first published in 2015 in Molecular Microbiology.

REPLY

The paragraph below was added to line 161 of the revised manuscript.

As mentioned already, spores are highly resistant to harsh conditions. This positive trait can also have negative effects on heterologous protein display with spores.  Electron and atomic force microscopy investigations demonstrated that stable macrostructures could form.  This arises from the self-assembly CotY, CotE, CotV and CotW to create one-dimensional fibers, two-dimensional sheets, and three-dimensional stacks.  This feature contributes to the inert spore property.  This stable structure may hinder the design of successful heterologous protein/spore coat protein fusions [19]

(4) In addition, the last sentence of the paragraph “There has been a recent investigation to identify suitable coat proteins for display fusions.” requires citation/s.

REPLY

This has been removed due to the deletion of the paragraph.
